# DNA Footprints: Using Parasites to Detect Elusive Animals, Proof of Principle in Hedgehogs

**DOI:** 10.3390/ani10081420

**Published:** 2020-08-14

**Authors:** Simon Allen, Carolyn Greig, Ben Rowson, Robin B. Gasser, Abdul Jabbar, Simone Morelli, Eric R. Morgan, Martyn Wood, Dan Forman

**Affiliations:** 1Gower Bird Hospital, Sandy Lane, Parkmill, Gower, Swansea SA3 2EW, UK; martyn@gowerbirdhospital.org.uk; 2School of Biological Sciences, University of Bristol, Bristol Life Sciences Building, 24, Tyndall Avenue, Bristol BS8 1TQ, UK; eric.morgan@qub.ac.uk; 3College of Science, Swansea University, Singleton Park, Swansea, Wales SA2 8PP, UK; c.greig@swansea.ac.uk (C.G.); d.w.forman@swansea.ac.uk (D.F.); 4Department of Natural Sciences, National Museum of Wales, Cardiff, Wales CF10 3NP, UK; ben.rowson@museumwales.ac.uk; 5Faculty of Veterinary and Agricultural Sciences, The University of Melbourne, Parkville 3010, Australia; robinbg@unimelb.edu.au (R.B.G.); jabbara@unimelb.edu.au (A.J.); 6Faculty of Veterinary Medicine, Teaching Veterinary Hospital, University of Teramo, 64100 Teramo, Italy; smorelli@unite.it; 7School of Biological Sciences, Queen’s University Belfast, 19 Chlorine Gardens, Belfast BT9 5DL, Northern Ireland, UK

**Keywords:** Hedgehog, PCR, *Crenosoma striatum*, rDNA, Gastropod, Nematode, Biological tag

## Abstract

**Simple Summary:**

Nocturnal and elusive animals are notoriously difficult to count—hedgehogs being a prime example. Therefore, any reliable way to demonstrate the presence of a particular animal, within a given area, would be a valuable addition to many ecologists’ tool kits. The proposed method is based upon the idea that you can find a parasite, specific to a vertebrate animal of interest that has a life stage within an invertebrate host. Molecular detection of these parasites is then carried out in the more abundant and easily collected invertebrate intermediate host. The key to this proposed method is the specificity of the parasite to the vertebrate animal and its detection in the invertebrate intermediate hosts. *Crenosoma striatum* is specific to hedgehogs and was chosen as the parasite to develop the molecular survey tool for hedgehogs, an elusive nocturnal species of considerable interest at present. Results revealed the presence of the nematode only at a site known to be inhabited by hedgehogs confirming the potential of this method to improve the accuracy of recording hedgehog populations.

**Abstract:**

The Western European Hedgehog (*Erinaceous europaeus*) is a nocturnal animal that is in decline in much of Europe, but the monitoring of this species is subjective, prone to error, and an inadequate basis for estimating population trends. Here, we report the use of *Crenosoma striatum*, a parasitic nematode specific to hedgehogs as definitive hosts, to detect hedgehog presence in the natural environment. This is achieved through collecting and sampling the parasites within their intermediate hosts, gastropoda, a group much simpler to locate and sample in both urban and rural habitats. *C. striatum* and *Crenosoma vulpis* were collected post-mortem from the lungs of hedgehogs and foxes, respectively. Slugs were collected in two sessions, during spring and autumn, from Skomer Island (*n* = 21), which is known to be free of hedgehogs (and foxes); and Pennard, Swansea (*n* = 42), known to have a healthy hedgehog population. The second internal transcribed spacer of parasite ribosomal DNA was used to develop a highly specific, novel, PCR based multiplex assay. *Crenosoma striatum* was found only at the site known to be inhabited by hedgehogs, at an average prevalence in gastropods of 10% in spring and autumn. The molecular test was highly specific: One mollusc was positive for both *C. striatum* and *C. vulpis*, and differentiation between the two nematode species was clear. This study demonstrates proof of principle for using detection of specific parasite DNA in easily sampled intermediate hosts to confirm the presence of an elusive nocturnal definitive host species. The approach has great potential as an adaptable, objective tool to supplement and support existing ecological survey methods.

## 1. Introduction

Objective methods for monitoring wild animals are needed to support management efforts, but are rarely straightforward, especially for elusive and nocturnal species. A complete census is usually impossible, and surveys more often rely on observations of individuals and indirect evidence of their presence, such as faecal counts or tracks [1]. With regards to elusive nocturnal animals specifically, even detection can be difficult, as exemplified by carnivore species that are widely dispersed, solitary and nocturnal [1,2,3]. Locating even the largest of terrestrial mammals, for example, the African forest elephant, can be a difficult task fraught with contestable results [4].

Western European Hedgehogs (*Erinaceus europaeus* Linnaeus, 1758) are classified as a species of least concern [5]; however, there is strong evidence of a recent decline in numbers across mainland Europe and in the UK [6,7,8,9,10]. Estimates suggest a reduction in UK populations within the range of 5–7% in the last 50 years [11], with one study suggesting a potential 25% reduction over the last decade [12]. Current survey methods rely on physical sightings and subjective evidence, such as scats (faecal deposits), tracks and carcases from road deaths, to determine the presence of hedgehogs [13,14,15,16]. Given the difficulties in sighting and correctly monitoring nocturnal animals, such as hedgehogs, there is a need to develop a wider panel of objective, evidence-based survey methods to supplement and confirm the findings of those currently used [17].

The use of parasites to monitor host populations has long been employed in the aquatic environment for fish populations [18,19,20,21], and more recently to quantify the presence of the elusive diamondback terrapin [22]. The use of parasites and their DNA as biological markers, however, remains underdeveloped in terrestrial environments. The parasitic nematode *Crenosoma striatum* is a lungworm highly specific to hedgehogs [23,24,25,26,27,28], and common in most populations. In a study of 74 dissected hedgehogs in the UK, 71% were found to be infected with *C. striatum* [29]. While hedgehogs are the sole definitive hosts for *C. striatum*, the available intermediate host range is much wider. Experimental infections comprising species from several gastropod (slug and snail) families of the orders Stylommatophora and Hygrophila [26,30] suggest a large number of potential hosts in hedgehog environments. Terrestrial molluscs are an integral part of many ecosystems and can be found across a diverse range of habitats throughout the British Isles [31,32,33].

It is here proposed that a polymerase chain reaction (PCR) based test could be used to rapidly and effectively determine the presence of *C. striatum* in local slug and snail populations, thereby indicating the presence or absence of hedgehogs within a given geographical area. If effective, this test would greatly facilitate monitoring of hedgehog distribution, and could potentially be adapted and developed for use in the monitoring of other species of interest. In the present study, this approach is evaluated by first devising a PCR assay specific for *C. striatum*, and then comparing results from areas of known hedgehog presence.

## 2. Materials and Methods

### 2.1. Isolation of DNA from Nematodes for Molecular Test Development

Adult worms of *C. striatum* were collected from the lungs of hedgehogs *post mortem*, and identified morphologically [29]. *Crenosoma vulpis*, a closely related species, collected from the lungs of red foxes (*Vulpes vulpes*) *post mortem*, was also used. DNA was extracted using the DNeasy Blood and Tissue Kit (Qiagen, Hilden, Germany); according to the manufacturer’s instructions, except that adult worms were initially ground in ATL buffer using a microfuge pestle. DNA was eluted in 100 μL and stored at −20 °C prior to analysis.

### 2.2. Primer Design and Multiplex Assay Development

The second internal transcribed spacer (ITS-2) of ribosomal DNA (rDNA) was chosen as the primary region of interest for primer design, due to its successful use in distinguishing between closely related nematodes in numerous previous studies [34,35,36,37,38,39,40]. To obtain sequence information for primer design, primer sequences NC1 and NC2 (Table 1, from Gasser et al. [41] 1993) were used to amplify the ITS-2 region of selected parasite DNA for sequencing. PCR conditions were optimised to achieve a single band of the expected size on an agarose gel. Specific products were purified by mini-column (Qiagen) and sequenced in both directions (Eurofins). Sequences obtained were aligned using the ClustalW function in BioEdit software [42], and a consensus sequence established for each species. Sequences from *C. striatum* (*n* = 2) and *C. vulpis* (*n* = 2) were compared with each other and with sequences from *Angiostrongylus vasorum* (a metastrongylid nematode using gastropod intermediate hosts and common in the study area [43], and *Aelustrongylus abstrusus* (a metastrongylid feline lungworm also using gastropod intermediate hosts). This was done to find suitable regions for the design of primers that would allow species differentiation by sequence and PCR product size (as illustrated in Appendix A). The ITS2 sequences of *Crenosoma* spp. Were submitted to GenBank with accession numbers MT808322 to MT808325. Primers were designed using Oligo6 (Molecular Biology Insights, Colorado Springs, CO, USA) to uniquely amplify a 157 bp region of *C. striatum* ITS-2 (C.St), and a 207 bp region of *C. vulpis* ITS-2 (C.Vu) (Table 1). Primers were checked with NCBI basic local alignment search tool (BLAST) for species specificity. An independent pair of primers for the amplification of a 710-bp fragment of the invertebrate mitochondrial cytochrome c oxidase subunit I gene (COX1) was selected [44] (henceforth termed COI) as a control to verify that DNA could be amplified from each sample. PCR conditions were optimised for both individual and multiplexed PCRs.

PCRs were performed in a volume of 15 µL including 2 µL of template DNA, 2.5 mM MgCl_2_, 0.2 mM dNTPs (Thermo Fisher, Loughborough, UK) 0.025 µ/µL GoTaq^®^ Flexi polymerase and 1× buffer (Promega, Southampton, UK) and 1× primer mix. 10× primer mixes were COI: 10 mM each primer, optimised multiplex 5 mM each C.St primer and 3 mM each C.Vu primer. The PCRs were carried out on a Biorad T100 Thermal Cycler using a touchdown profile, consisting of an initial denaturation at 95 °C for 3 min followed by nine cycles of 94 °C for 30 s, 65 °C (1 °C decrease per cycle) for 20 s, 72 °C extension then 33 cycles of 94 °C for 30 s 55 °C for 20 s and 72 °C extension. Extension at 72 °C was for 30 s for the multiplex PCR and 1 min for the COI PCR. The final extension was 10 min at 72 °C. PCR products were examined on 1% agarose gels stained with GelRed™ (Biotium Inc., Fremont, CA, USA).

The multiplex PCR was initially checked for analytical specificity by testing against a species panel of DNA isolated from morphologically identified adult lungworms, and confirmed to be diagnostic for C.St and C.Vu (see Appendix A).

For PCR testing of slug DNA, an initial control COI PCR was performed prior to the test C.St-C.Vu multiplex, and was negative for some samples, mostly from *Arion ater* slugs, and some appeared tinged with a dark colour. For these, 2 μL of genomic DNA was examined on an agarose gel, and the presence of high molecular weight DNA in the extraction was confirmed. Attempts were made to re-purify to negate the effects of inhibitors. For most samples, PCR was successful with the addition of PCRboost^®^ (Biomatrica, San Diego, CA, USA). Multiplex PCRs were carried out under the same conditions for these samples.

PCRs were repeated twice to verify an amplification (test positivity). Test-negative PCRs were scored only if samples with a positive PCR for the control invertebrate COI PCR. The results of the C.St-C.Vu multiplex on positive COI PCR’s were analysed using an exact binomial test. 

### 2.3. Slug Samples

In order to demonstrate the correlation between *C. striatum* incidence and the presence of hedgehogs, slugs were collected in autumn from Skomer Island, covering an area of approximately 160 ha, and in both spring and autumn in Pennard, covering an area of 0.36 ha: Both areas are in south-west Wales, UK. There are no known reports of hedgehogs (or Foxes) on Skomer Island (personal communication with Mark Hodgson, Wildlife Trust South West Wales), whereas Pennard is an area with an abundant local hedgehog population; more than 180 individuals from this particular region were admitted to Gower Bird Hospital wildlife rehabilitation centre between 2001 and 2017. The slugs collected were identified morphologically by BR^4^ and then stored at −20 °C before processing. The posterior foot section of each slug was removed and macerated prior to tissue lysis.

### 2.4. Gastropod DNA Extraction

Genomic DNAs from 80 slugs were extracted from slug tissue using Dneasy Blood and Tissue Kits (Qiagen, Hilden, Germany) employing a Maxwell^®^ 16 MDx Research System (Promega, Maddison, WI, USA) as recommended by the manufacturers. Any undigested tissue and pigment from the larger *Arion ater* specimens were removed by centrifugation before spin column purification. DNA was eluted in 100 μL and stored at −20 °C prior to further analysis.

### 2.5. Sample Size Calculator

The number of slugs required to be sampled to provide a reliable indicator of the absence of *C. striatum* infection, and hence, the absence of hedgehogs, was simulated using the binomial distribution. Thus, the required sample size was defined as that yielding a <0.05 probability of zero successes (=detected infections), at a given above-zero true prevalence (p. 64, [45]). This is the sample size needed to avoid a type II error, i.e., falsely declaring the absence of *C. striatum* when actually present, at *p* = 0.05.

## 3. Results

Out of the 80 slugs 17 were excluded (Table 2), due to negative COI result. Slug samples from Pennard collected in spring (*n* = 20) and autumn (*n* = 22) represented nine species. Overall, the prevalence of *C. striatum* in this sample set was 10% (95% exact binomial confidence bounds 3–23%). Species infected with *C. striatum* were *Arion subfuscus* (spring; *n* = 1), *Arion ater* agg. (autumn; *n* = 1) and *Tandonia sowerbyi* (autumn, *n* = 2). Additionally, the *A. ater* agg. Individual was concurrently infected with *C. vulpis*, confirming the sensitivity of the assay without cross-species amplification. The Skomer slug samples collected in autumn (*n* = 21) comprised two species: *A. ater* and *Lehmannia marginata*. Neither *C. striatum* nor *C. vulpis* was detected in any of these samples. Results of the sample size simulation are presented in Figure 1. At the 10% prevalence observed in this study, a sample of 29 slugs would be needed to reasonably (at *p* = 0.05) avoid a false negative, i.e., erroneously conclude that infection is absent. The number of slugs needed would rise at lower prevalence, and fall at higher prevalence.

## 4. Discussion

This study demonstrates the use of a multiplex test for *Crenosoma* species, which can accurately identify and discriminate between closely related species *C. striatum* and *C. vulpis* from slug tissues. The fact that no *C. striatum* was detected in the Skomer sample set indicates the potential of *C. striatum* as an indicator species for the presence of elusive hedgehogs in any given locale. Furthermore, the sensitivity of the assay suggests that other parasites highly specific to host species of interest could be used in this way for monitoring and surveillance, for instance as part of management programmes for endangered or invasive species [4,46]. Direct detection of environmental DNA also has potential for monitoring of elusive species [47,48]. Detection of host-specific parasites within intermediate hosts, as proposed here, has the advantages of focusing sampling and potentially longer persistence of DNA in the form of living immature parasite stages.

The methodology described here may need refinement in terms of sample preparation. Some parasites have a preferred site within their host; for instance, *Angiostrongylus vasorum* occupies the right ventricle and pulmonary arteries in its vertebrate hosts [49], whilst *C. striatum* prefers the bronchioles and bronchi of the lungs [26]. The affinity of these parasites to particular sites within the host may extend to the intermediate host, such that sub-sampling of tissue could bias results and affect method sensitivity. Further research needs to be carried out to determine if *C. striatum* has a predilection site in slugs, to increase the efficacy of detection in slug tissue.

To increase the chances of detecting a parasitised slug, species that have been active the longest, and therefore, had the greatest opportunity to acquire parasite infections should, in principle, be targeted for sampling. For example, *A. subfuscus* activity has been seen to peak between May and June with little between-year deviation [50], making it an ideal candidate for spring and summer sampling. The present study found *A. subfuscus* to be the only species with a positive *C. striatum* result in spring sampling. Similarly, *A. ater* and *T. sowerbyi* would be of major interest in autumn and winter sampling, with their peak activity being in January or between August and October, respectively [50]. *Arion ater* may be of particular interest in future research, as it was the only species that presented simultaneous infection with both *C. striatum* and *C. vulpis*. Additionally, the detection of *C. striatum* in *A. ater*, *A. subfuscus* and *T. sowerbyi* appears to be the first confirmed report of infection in these species [30]. This suggests that the potential intermediate host range of *C. striatum* could be much greater than previously thought. Extensions to the present study could further develop the test for hedgehog monitoring through targeting particular slug species and anatomical sites, and by matching the target sample size to the expected prevalence and required precision. The number and cost of PCR assays performed per geographical site could also be reduced by pooling samples from different slugs. These refinements require validation and could establish whether parasite abundance in slugs is related to hedgehog population density, which if it were found to be the case, would enhance its utility as a monitoring tool. Regardless of this relationship, however, results here suggest that presence or absence of *C. striatum* correlates, as expected, with that of its hedgehog definitive host, and can, therefore, be used as a robust indirect indicator of hedgehog presence. The required number of slugs to be sampled in order to reasonably exclude the possibility of *C. striatum* depends on the underlying prevalence, which is unlikely to be known in a newly surveyed site. Further information on the range of prevalence of *C. striatum* infection in gastropods in areas inhabited by hedgehogs would, therefore, be useful to evaluate the feasibility and efficiency of the present approach across the species range.

The approach presented here could be extended to other systems, where highly host-specific parasites are present at reasonably high prevalence, distinguishable from closely related species, and accessible, for example in easily sampled intermediate hosts. The most fundamental of these factors is the host specificity. Host specificity is often under or over-estimated for parasitic species [51], and parasite-host interactions are rarely well-understood in wild animals [52]. Most parasites can infect multiple host species [53,54,55], albeit to a highly varied extent [56], rendering most as unsuitable for host population studies. Helminths, however, often demonstrate high host-specificity, with nearly 50% of those reported in one study of primates inhabiting a single host species [54]. The sensitivity of the assay presented herein demonstrates that quick and accurate delineation between closely related parasite species can be achieved. It is entirely possible that this methodology could be adapted to other vertebrate species of conservation concern, wherever a suitable parasite species can be identified. To date, only a small number of parasites with singular definitive hosts have been described; Table 3 provides examples of such species. It may be the case that host-specific helminths occur commonly; however, further research is needed in order to clarify this. Furthermore, taxonomic revision frequently leads to a reassessment of host specificity: For example, many nematodes found in amphibia had been previously identified as *Rhabdias rana*, molecular analysis later demonstrated historical misidentification [57], and new species were described as a result. Therefore, it is quite possible that many parasitic species identified before the modern molecular biology era, may have been incorrectly described, increasing the possibility of detecting species-specific and molecularly distinct parasites with potential as indicators of host presence. In addition to taxonomy, ecological factors determine the realisation of potential host range, and are changing in many systems [58]. Shifts in prevalence and host range might have to be taken into account during parasite-based monitoring programmes, and at the same time can provide additional information on host ecology and infection patterns.

Further improvements could be made through development as a loop-mediated isothermal amplification (LAMP-PCR), using similar methodology to that previously described [59,60]. This has potential for a test which could be used in a field setting: Feng et al. [59] found the LAMP-PCR method had lower, but adequate sensitivity for the specific detection of cestode DNA as compared to multiplex PCR, while Abbasi et al. [60] demonstrated 10-fold increased sensitivity over PCR for the detection of *Schistosoma* spp. in infected snails.

**Table 3 animals-10-01420-t003:** Examples of helminth species that to date, have only been identified in a singular definitive host.

Parasite Group	Helminth Species	Definitive Host	Common Name	Host Class	Reference
Nematode					
	*Abbreviata perenticola*	*Varanus giganteus*	Perentie	Reptilia	[61]
	*Abbreviata physignathi*	*Physignathus lesueurii*	Australian Water Dragon	Reptilia	[61]
	*Abbreviata glebopalmae*	*Varanus glebopalma*	Black Palmed Rock Monitor	Reptilia	[61]
	*Abbreviata barrowi*	*Pseudechis australis*	Mulga Snake	Reptilia	[61]
	*Nematodirus davtiani alpinus **	*Capra ibex*	Alpine Ibex	Mammalia	[62]
	*Filaroides martes*	*Martes americana*	American Pine Marten	Mammalia	[63]
	*Perostrongylus falciformis*	*Meles meles*	European badger	Mammalia	[64]
	*Crenosoma striatum **	*Erinaceus europaeus*	European Hedgehog	Mammalia	
	*Rhabdias bakeri*	*Rana sylvatica*	Wood Frog	Amphibia	[57]
	*Rhabdias ambystomae*	*Ambystoma maculatum*	Spotted Salamander	Amphibia	[65]
	*Parachordatortilis mathevossianae*	*Falco tinnunculus*	Common Kestrel	Aves	[66]
	*Physaloptera apivori*	*Pernis apivorus*	European Honey Buzzard	Aves	[66]
	*Physaloptera Mexicana **	*Buteo buteo*	Common Buzzard	Aves	[66]
	*Serratospiculum tendo*	*Falco peregrinus*	Peregrine falcon	Aves	[66]
Trematode					
	*Urotrema scabridum **	*Anolis sagrei*	Brown Anole	Reptilia	[67]
	*Pleurogonius malaclemys*	*Malaclemys terrapin*	Diamondback Terrapin	Reptilia	[22]
	*Mesocoelium lanfrediae*	*Rhinella marina*	Cane Toad	Amphibia	[68]
	*Parastrigea intermedia*	*Circus aeruginosus*	Western Marsh Harrier	Aves	[66]
Cestode					
	*Schistotaenia tenuicirrus*	*Podiceps grisegena*	Red Necked Grebe	Aves	[69]
	*Cladotaenia foxi*	*Falco Peregrinus*	Peregrine falcon	Aves	[66]

* denotes parasite species for which the definitive host is geographically isolated from other host species.

## 5. Conclusions

We conclude that proof of principle has been demonstrated in using terrestrial parasite DNA to confirm the presence of hedgehogs in a given locale. PCR tests can be used to effectively detect and delineate isolates of *C. striatum* and *C. vulpis* from gastropod samples. A critical assessment of different slug tissue and nematode extraction methods, and epidemiological factors, is necessary for the improvement and development of the method described here. This method could provide significant support for monitoring and conservation efforts in hedgehogs, and could pave the way for similar methods to be employed for monitoring of other terrestrial species whose conservation is of concern.

## Figures and Tables

**Figure 1 animals-10-01420-f001:**
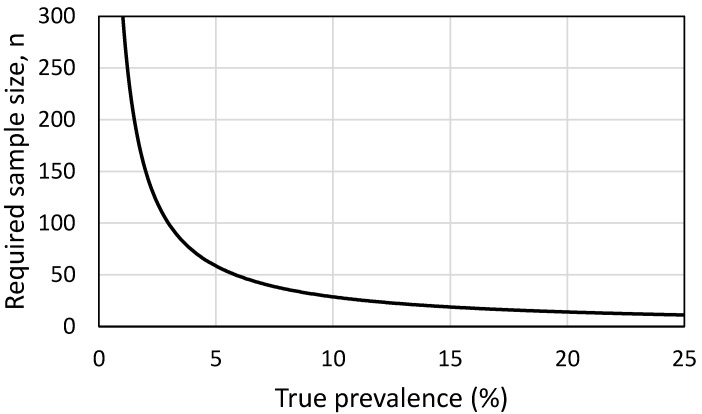
The sample size (=number of slugs) required to detect at least one infected slug, given true prevalence from 1% (*n* = 299) to 25% (*n* = 11). Higher prevalence omitted for clarity: *n* declines further to 5 (at 50% prevalence) and 3 (75%).

**Table 1 animals-10-01420-t001:** The primers and their loci used in PCR tests to identify the presence or absence of *Crenosoma striatum* in slugs.

Primer	Sequence (5′-3′)	Locus	References
**NC1**	ACGTCTGGTTCAGGGTTGTT	5.8S nuclear rRNA gene	[41]
**NC2**	TTAGTTTCTTTTCCTCCGCT	28S nuclear rRNA gene	[41]
**C.St-ITS2F**	CGATTCCCGTTCTAGTTGAGAC	ITS-2 nuclear rDNA	this study
**C.St-ITS2R**	AAAACCACCTCGACGACATC	ITS-2 nuclear rDNA	this study
**C.Vu-ITS2F**	CGATTCCCGTTTTAGTTAAGGA	ITS-2 nuclear rDNA	this study
**C.Vu-ITS2R**	GCTTATCAATCGTCGAATATCATGC	ITS-2 nuclear rDNA	this study
**LCO1490**	GGTCAACAAATCATAAAGATATTGG	Mitochondrial *cox*1 gene	[44]
**HC02198**	TAAACTTCAGGGTGACCAAAAAATCA	Mitochondrial *cox*1 gene	[44]

C.St-ITS2F/R and C.Vu-ITS2F/R = forward and reverse primers for *Crenosoma striatum* and *Crenosoma vulpis*.

**Table 2 animals-10-01420-t002:** All slug samples included in the proof of principle with their nucleic purity, tissue weight and COI result.

		**DNA**	**260/280**	**Weight**		
**Sample ID**	**Slug Species**	**μg/mL**	**Ratio**	**mg**	**COI**	**Excluded**
**Pennard Spring**					
1	*Tandonia sowerbyi*	59.12	1.54	50	+ve	
2	*Tandonia sowerbyi*	86.36	1.60	34	+ve	
3	*Tandonia sowerbyi*	45.64	1.60	40	+ve	
4	*Tandonia sowerbyi*	102.74	1.83	47	−ve	x
5	*Deroceras panormitanum*	151.01	1.91	42	−ve	x
6	*Deroceras panormitanum*	179.24	2.00	63	+ve	
7	*Deroceras panormitanum*	180.09	1.97	55	−ve	x
8	*Deroceras panormitanum*	94.04	1.79	48	−ve	x
9	*Lehmannia marginata*	7.19	1.37	50	+ve	
10	*Lehmannia marginata*	13.98	1.42	56	+ve	
11	*Lehmannia marginata*	108.34	1.77	42	−ve	x
12	*Lehmannia marginata*	54.50	1.89	48	+ve	
13	*Arion hortensis agg.*	30.28	1.55	45	+ve	
14	*Arion hortensis agg.*	37.04	1.70	41	+ve	
15	*Arion hortensis agg.*	25.47	1.63	38	+ve	
16	*Arion hortensis agg.*	26.16	1.49	50	−ve	x
SL-1	*Tandonia sowerbyi*	32.00	2.05	200	+ve	
SL-2	*Tandonia sowerbyi*	325.69	2.03	141	+ve	
SL-3	*Tandonia sowerbyi*	310.11	2.04		+ve	
SL-4	*Arion subfuscus*	464.97	2.00		+ve	
SL-5	*Arion subfuscus*	300.12	2.04		−ve	x
SL-6	*Arion subfuscus*	73.10	1.84		−ve	x
SL-7	*Arion subfuscus*	173.21	1.98		−ve	x
SL-8	*Arion subfuscus*	128.40	1.94		−ve	x
SL-9	*Arion flagellus*	45.35	1.89		+ve	
SL-10	*Arion flagellus*	226.78	2.02		+ve	
SL-11	*Arion flagellus*	153.25	1.90		+ve	
SL-12	*Arion flagellus*	107.87	1.89		+ve	
SL-13	*Arion flagellus*	107.57	1.89		+ve	
SL-14	*Arion flagellus*	200.06	1.99		+ve	
**Pennard Autumn**					
11.1	*Arion ater agg.*	27.62	1.46	60	+ve	
11.2	*Arion ater agg.*	46.00	1.81	50	+ve	
11.3	*Arion ater agg.*	23.00	1.61	50	+ve	
11.4	*Arion ater agg.*	57.00	1.71	50	+ve	
28.1	*Arion ater agg.*	107.77	1.70	50	−ve	x
28.2	*Arion ater agg.*	44.54	1.68	50	+ve	
28.3	*Arion ater agg.*	31.04	1.41	50	−ve	x
28.4	*Arion ater agg.*	44.49	1.61	50	+ve	
15.1	*Arion ater agg.*	32.01	1.49	50	+ve	
15.2	*Arion ater agg.*	62.77	1.67	50	+ve	
15.3	*Arion ater agg.*	10.13	1.12	50	+ve	
15.4	*Arion ater agg.*	61.56	1.80	80	+ve	
2.1	*Limax flavus*	0.00	−0.69	50	−ve	x
2.2	*Limax flavus*	17.91	1.26	50	+ve	
2.3	*Limax flavus*	6.27	1.12	60	+ve	
2.4	*Limax flavus*	93.23	1.77	60	+ve	
1.1	*Tandonia sowerbyi*	76.78	0.92	58	+ve	
1.2	*Tandonia sowerbyi*	43.57	1.57	54	+ve	
1.3	*Tandonia sowerbyi*	56.92	1.70	44	+ve	
1.4	*Tandonia sowerbyi*	40.62	1.49	47	+ve	
22.1	*Arion rufus*	2.55	0.72	58	+ve	
22.2	*Arion rufus*	0.59	0.61	48	−ve	x
22.3	*Arion rufus*	19.40	1.21	41	+ve	
22.4	*Arion rufus*	11.28	1.06	52	+ve	
23.1	*Arion rufus*	7.08	1.08	49	+ve	
23.2	*Arion rufus*	25.60	1.34	57	+ve	
**Skomer Autumn**					
SK1	*Arion ater*	179.7	1.8	30	+ve	
SK2	*Arion ater*	114	1.85		+ve	
SK3	*Arion ater*	53.2	1.87		+ve	
SK4	*Arion ater*	105.1	1.83		+ve	
SK5	*Arion ater*	129.1	1.85	60	+ve	
SK6	*Arion ater*	172.3	1.75		+ve	
SK7	*Arion ater*	160.3	1.64	100	−ve	x
SK8	*Arion ater*	271.4	1.97	20	−ve	x
SK9	*Arion ater*	266.8	1.93	41	−ve	x
SK10	*Arion ater*	118.8	1.86		+ve	
SK11	*Lehmannia marginata*	107.4	1.86	48	+ve	
SK12	*Lehmannia marginata*	150.8	1.95		+ve	
SK13	*Lehmannia marginata*	174.7	1.93	19	+ve	
SK14	*Lehmannia marginata*	166.4	2.02		+ve	
SK15	*Lehmannia marginata*	161.3	1.93		+ve	
SK16	*Lehmannia marginata*	279.7	1.95		+ve	
SK17	*Lehmannia marginata*	355.2	1.95		+ve	
SK18	*Arion ater*	100.1	1.91		+ve	
SK19	*Arion ater*	73.1	1.82		+ve	
SK20	*Arion ater*	97.2	1.87	83	+ve	
SK21	*Arion ater*	108.7	2.1		+ve	
SK22	*Arion ater*	233.4	1.92		+ve	
SK23	*Arion ater*	130.9	1.99		+ve	
SK24	*Arion ater*	130.6	1.9		+ve

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
