# Peer review of "DNA Footprints: Using Parasites to Detect Elusive Animals, Proof of Principle in Hedgehogs"

_animals, 2020, doi:10.3390/ani10081420_

Round 1
Reviewer 1 Report
General comments: The study tested the use of the detection of host-specific parasite (C. striatum) in a more abundant intermediate hosts to confirm the presence/absence of Western European Hedgehogs. The authors concluded that this method could be used to assess the presence of the hedgehog, because C. Striatum were only found in nematodes in the site where hedgehogs occurred. The concept idea is novel and interesting. The manuscript is clearly described. The molecular tool described in the study can be adapted to aids in ecological surveys of hedgehogs or other elusive species.
There are a couple points that the authors should address: did the authors conduct a control experiment to confirm that the developed primers are specific to C. Striatum and C. vulpis (i.e., demonstrating that the primers can detect the presence or absence or the parasites in known specimens)? and if so, what was the level of accuracy?
Reviewer 2 Report
The study investigates the use of parasite identification to investigate host populations in mammals, using the hedgehog as an example. The approach is intriguing but the study is presented very minimalistic, the main conclusion has to be toned down because the study design allows for a proof of concept but not in the sense that allow to “ confirm the presence or absence of hedgehogs in a given locale” (conclusion).
These conclusions can not be drawn because the populations used for the study are not informative for all aspects that are needed to use “detection of specific parasite DNA in easily sampled intermediate hosts to confirm the presence of an elusive nocturnal definitive host species”. Only two locations had been looked at, which were also very different in size as 160 and 0.36 ha (if that was not an error). The large locality is an island where hedgehogs might have never occurred. The result does show therefore that if the species has been always absent in an certain area the parasite is missing. It does not show if the occurrence of the parasite always correlates with the presence of the host. It is reasonable to assume that vanishing of the host will lead to vanishing of the parasite but only after a certain timelag, which might be very long
The text has to make clear that the study provides only a single observation of a locality where no slug could be found infected and one where slugs could be found and this correlates with long term presence of hedgehog. This is suitable as a proof of concept, the necessary design to verify correlation of parasite and hedgehog population dynamics has to be discussed. To go from concept to potential implementation. occurrence patterns of the parasite could also be informative for other aspects of population dynamics then only occurrence data.
The verification of the specifity of the pcr product by sequencing is missing, or I could not find one. The design uses primer as species discriminating tool. Even if the close related parasite for foxes can be excluded, other, not yet known closely related species could be detected which would provide false positives in the sampling. The short comings of species delimitation and uncertainy in host specifity plays a role here and had been discussed in the study.
The simulation is not covered well enough in the material and methods. The citation 45 page 64 is an online document which is not easily found on the page indicated. Assessment date is missing. The simulation results in 29 individuals to exclude the possibility of false negative locations, but is met by 21 samples from skomer island. I wonder why the simulation is done when it does not have a consequence on the study design.
Regardless, the problem might rather be the possibility of false negatve slug samples which can arise from a number of factors, like number of infections in the slug. It might be that only after a certain number of individuals of the parasite that can be found in the slug the pcr is positive. The species of slug might play a role, slugs are notoriously difficult to isolate pcr quality DNA from, and so on. This is influencing the prevalence (as variable in the simulation) and might underestimate the slug infection rate and increase variation between localities. The authors developed for this probably the multiplex assay, the results of this multiplex are not really presented, though. Only some notes in the material and methods section. All results should be included in the study though, also negative ones.
The results could be presented in a table, of which species and how many individuals per locus and how many were positive. For example one slug species had two infections in autumn, how many of this species are in the autumn sampling? Here the amount of positive and negative control DNAs could be included.
The prevalence of about 10 % results obviously form the number of all individuals collected in Pennard. However, because on the island only one sampling in autumn was done, the two pennard samplings constitute replicates and should be treated not additive. The sample for comparison with the island would be the pennard autumn sampling.
The differences in slug species composition between the localities might be a strong confounding factor and what would be needed in the end is an estimate for infection rate of different slug species using the method. The species difference should be discussed more thoroughly. Like pointed out below in discussion line 2015ff the study design is questionable in this point. Is there a reason to apriori assume that Lehmannia marginata is able to serve as intermediate host?
Line 100 ff, are the sequences provided in genbank?
Line 110ff and 116ff: the multiplex assay is not well characterized. It remains also unclear if the results reported stem from a multiplex or single plex assay.
Line 131 in this sense, the number of negative DNA amplifications have to be reported. Like the study and data are reported it is also possible that the reported 60 samples are only a small subset of COI positive samples from a larger study. All samples have to be reported, also the ones that are later excluded. Best as a table summarizing these results.
Line 132: The statistics used is not described well enough and also not the rationale behind it. In line 132 “ All PCR data were analysed using exact binomial and Chi-squared tests using R [45]” What is tested here? What are PCR data? All tests and their outcome have to be presented in the result section (see the comment there).
Line 136 Material method should include how many samples had been collected per site and time. The information is given in the result section.
Line 160 ff The results have to contain all aspects of sampling, negative and positive pcrs, species composition per location and so on. Most points are indicated above.
Line 164 statistics. I am do not see how the statistics is meaningful here. There is no replication. When the chi square test does not indicate significance between one positive in 20 samples and 3 in 22, is 0 in 20 significant different from this? If not, how the conclusions of the study can be drawn?
Line 179 delineation between the parasite species: this has not been shown. The only indication is that in one individual both species had been amplified. Because striatum amplicon is a subset of vulpis amplicon it can be that in case vulpis is amplified, striatum amplification is facilitated from the vulpis amplicon in a multiplex. Comparison of the primers in table 1 show that foreward primer is designed on the same position, but reverse primer not. I assume that design had been done in this way to discriminate conveniently using length information. Sequencing of the positive samples for verification would help. Adjust also line 204 ff
Line 205 to 206: the statement of first time detection of the parasite in the species allows to question the overall design of the study. It looks like that for most of the included samples an infection could not have been expected. It might be also that even if the intermediate host persist in these species, infections are only occasionally. The species known to be main intermediate host should be used to determine prevalence, the other species can than be used to test if the same prevalence can be found. Then all the species would be equally able to carry the parasite. This has to be discussed and maybe also the questions adjusted respectively.
Line 215 to 2016. Above I commented already about conclusions in this direction. I would like to add here, that eggs of helmints are known to exceptionally long persist in the environment. In addition they can be transported over far distances. If this is the case should be included in the discussion. And also what it means to determine population dynamics of hedgehogs. So far it looks like only areas where the species was not existing for a long time show negative results.
